# Medical Application of Substances Derived from Non-Pathogenic Fungi *Aspergillus oryzae* and *A. luchuensis*-Containing *Koji*

**DOI:** 10.3390/jof7040243

**Published:** 2021-03-24

**Authors:** Hiroshi Kitagaki

**Affiliations:** Graduate School of Advanced Health Sciences, Saga University, Saga 840-8502, Japan; ktgkhrs@cc.saga-u.ac.jp

**Keywords:** medical application, *koji*, fermentation, non-pathogenic fungus, medicinal substances

## Abstract

Although most fungi cause pathogenicity toward human beings, dynasties of the East Asian region have domesticated and utilized specific fungi for medical applications. The Japanese dynasty and nation have domesticated and utilized *koji* fermented with non-pathogenic fungus *Aspergillus oryzae* for more than 1300 years. Recent research has elucidated that *koji* contains medicinal substances such as Taka-diastase, acid protease, *koji* glycosylceramide, *kojic* acid, oligosaccharides, ethyl-α-d-glucoside, ferulic acid, ergothioneine, pyroglutamyl leucine, pyranonigrin A, resistant proteins, deferriferrichrysin, polyamines, *Bifidobacterium*-stimulating peptides, angiotensin I-converting enzyme inhibitor peptides, 14-dehydroergosterol, beta-glucan, biotin, and citric acid. This review introduces potential medical applications of such medicinal substances to hyperlipidemia, diabetes, hypertension, cardiovascular and cognitive diseases, chronic inflammation, epidermal permeability barrier disruption, coronavirus disease 2019 (COVID-19), and anti-cancer therapy.

## 1. Introduction

Fungi in general cause many infectious pathogenicities in human beings, including *Aspergillus*, *Candida*, *Cryptococcus*, and *Mucor* species [1]. However, several fungi have been utilized as medicines across the world since the prehistoric ages. Fungi are eukaryotic microbes that have complex biosynthesis pathways and various morphologies. They have unique biosynthesis pathways and thus produce unique substances that are not contained in mammals, plants, or prokaryotic bacteria [2].

Within fungi, mushrooms have attracted attention in terms of their medical potential since the prehistoric ages [3], possibly because of their visibility without microscopes. For example, hoelen, which is a *Wolfiporia cocos* mushroom proliferating on the roots of pine trees, has been used in East Asian countries, including China and Japan, as a medicine for more than 1800 years [4]. Moreover, a *Ganoderma lucidum* mushroom is described in Shennong Ben Cao Jing, which was written in 25–220 A.D. in China, as being able to increase the lifespan [5].

The beta-glucan contained in *Schizophyllum commune*, Schizophyllan, and *Lentinula edodes*, Lentinan, has been registered and utilized as immunoactivating medicines to treat cervical, stomach, and colon cancer [6]. However, medical applications of fungi which do not form mushrooms had not been described until the modern age, possibly because they are invisible without microscopes and microscopes only prevailed across the world in the 18–19th centuries.

The use of a red fungus *Monascus* in China is one of these examples [7]. The health benefits of red rice *koji* has been recognized in China for more than approximately 700 years. Its medical use was first described in Compendium of Materia Medica, which was written in 1498 A.D., indicating that it clears the blood and supports digestion of foods in the intestine. Dr. Endo elucidated, in 1979, that monacolin K produced by *Monascus ruber* has a hypocholesterolemic effect [8]. Moreover, it also contains medicinal substances such as monascin [9], ankaflavin [10], and ergosterol [11].

Moreover, the fungus *Penicillium camemberti* or *roqueforti*, belonging to ascomycetes, has been used to manufacture blue cheese in France for centuries. In 1929, Sir Fleming found that a fungus, *Penicillium rubens*, kills pathogenic bacteria [12]. Later, penicillin, which was identified as a substance that kills pathogenic bacteria, and related antibiotics were utilized to eradicate bacterial infectious diseases across the world. 

Tempeh, which is an Indonesian traditional fermented food lasting for more than 400 years, and produced by fermenting soybeans with *Rhizopus oligosporus* or *R. oryzae*, has several health benefits such as improvement of cognitive function, gut immunity, intestinal microbial flora, hyperlipidemia, and anemia [13].

Since the earliest historical age of Japan, Japanese people have isolated fungus *Aspergillus oryzae*, belonging to ascomycetes, and domesticated and utilized it to produce *koji* as a catalyst of starch in rice. During the process that lasted for 1300 years, non-pathogenic strains were genetically selected by specialized technicians of *koji*. Indeed, miso is described in Taihoritsuryo, which is one of the earliest legal codes announced by the emperor Monmu in 701, A.D. (Figure 1). The *koji* drink amazake is described in Manyoshu, which is a collection of poems of the imperial court written in 731–733 A.D. (Figure 1). A battle between the specialized technicians and the Muromachi government is described in 1444, A.D. (Koji Battle of Bunn-an). Therefore, it is evident from the literature that specialized technicians have genetically selected and maintained the strains, and *A. oryzae* strain presently used to produce *koji* is considered to be non-pathogenic. Ryukyu people in Okinawa Islands also domesticated *Aspergillus luchuensis*. Therefore, *koji* that uses these fungi are generally regarded as safe (GRAS) by the FDA and is used in most Japanese fermented foods and drinks, including miso, soy sauce, sake, amazake, vinegar, kurozu, fermented barley extract, and shochu as saccharifyer of starch contained in steamed rice, barley, or soybeans [14]. Japanese people have long recognized the health benefits of these fermented foods. Indeed, Honcho-shokkan describing the medical benefits of foods in Japan written in 1695, A.D. describes that dietary intake of *koji* supports the digestion of foods, decreases occlusion, and smoothens the blood flow. Yojo-kun (regimen sanitatis for longevity of Japan), written by the Japanese scholar Ekken Kaibara in 1712, A.D. mentions that the dietary intake of miso is gentle for the body and compensates for the functions of the gastrointestinal tract. Furthermore, the average life span of Japanese people is one of the longest in the world, and Japanese cuisines contain many fermented foods, most of which contain *koji* [15]. It could thus be hypothesized that the substances contained in Japanese fermented foods and/or *koji* are beneficial for health. However, medical knowledge on this non-pathogenic fungus, *A. oryzae*, is possibly limited because of its modest medicinal effect and is still under study at present.

Recent studies in Japan have elucidated that *koji* contains many substances that have potential medical applications, and this review summarizes such studies. This novel knowledge will pave the way for medical applications of non-pathogenic fungus *A. oryzae* in the future. 

## 2. Taka-Diastase and Acid Protease

A Japanese scholar, Jokichi Takamine, applied a patent describing a starch-degrading enzyme preparation derived from *koji* and designated it as Taka-diastase in 1911 [16]. It has been widely used as a starch digestant to treat stomach upset, stomachache, heartburn, and overeating throughout the world.

Recently, the dietary intake of the acid proteases contained in *Aspergillus oryzae* or *A. luchuensis* was shown to increase intestinal *Bifidobacterium* and *Lactobacillus* spp. [17,18]. This could be attributed to the remnant of enzymatic activity when they pass through the stomach and the small intestine. Therefore, acid protease could be a new generation of prebiotics.

## 3. *Koji* Glycosylceramide

Glycosylceramide is a sphingolipid consisting of a sphingoid base, fatty acid, and a sugar moiety (Figure 2). It is contained in various species, including plants and fungi. Sphingolipids have versatile biological functions, such as signal transduction [19], enhancement of the immune system [20], and raft formation [21]. Their structures have great variations, depending on the structure and type of the sphingoid base, fatty acid, and sugar moiety. The characteristic structure of *A. oryzae* is *N*-2′-hydroxyoctadecanoyl-l-*O*-β-d-glucopyranosyl-9-methyl-4,8-sphingadienine, *N*-2′-hydroxyoctadecanoyl-l-*O*-β-d-glucopyranosyl-4,8-sphingadienine, *N*-2′-hydroxy-3′-octadecenoyl-l-*O*-β-d-glucopyranosyl-9-methyl-4,8-sphingadienine, *N*-2′-hydroxyhexadecanoyl-l-*O*-β-d-glucopyranosyl-4,8-sphingadienine, *N*-2′-hydroxyicosanoyl-l-*O*-β-d-glucopyranosyl-4,8-sphingadienine, *N*-2′-hydroxyicosanoyl-l-*O*-β-d-glucopyranosyl-4,8-sphingadienine, and the corresponding molecules in which sugar moiety is substituted for galactose instead of glucose [22,23,24,25,26]. It is contained in *koji* and Japanese fermented foods at 1–4 mg/serving [27]. Most of the glycosylceramide in *koji* is considered not to be degraded by enzymes in the small intestine, and passes through to the large intestine [28]. The dietary intake of the glycosylceramide purified from *koji* at a concentration of 1% *w*/*w* of diet in mice increases intestinal *Blautia coccoides*, *Bacteroides sartorii*, and *Hathewaya histolytica* in mice. Since *B. coccoides* is considered to be a beneficial microbe, *koji* glycosylceramide could be considered as a new type of prebiotics. Also, feeding of glycosylceramide at a concentration of 1% *w*/*w* of diet to obese mice increases bile acids and lowers liver cholesterol in obese mice [29]. Since liver cholesterol is one of the causes of hyperlipidemia, *koji* glycosylceramide could potentially be medically utilized to treat hyperlipidemia.

Furthermore, glycosylceramide purified from *koji* increases the expression of the genes involved in tight junctions and ceramide delivery in normal human epidermal keratinocytes [30]. The dietary intake of the glucosylceramide derived from konjac extract improves transepidermal water loss (TEWL) in hairless mice with sodium dodecyl sulfate (SDS)-induced skin roughness [31]. Since the glycosylceramide contained in *koji* contains the same species of glucosylceramide contained in konjac [23], the dietary intake of *koji* might improve TEWL. Therefore, the dietary intake of *koji* or the topical administration of *koji* to the skin might be beneficial for skin disorders.

## 4. Kojic Acid

*A. oryzae* produces kojic acid (Figure 3), which functions as a competitive and reversible inhibitor of animal and plant polyphenol oxidases, xanthine oxidase, and some amino acid oxidases, and is used as an food additive for preventing enzymatic browning or in cosmetic preparations as a skin-lightening agent [32]. Therefore, *koji* contains kojic acid at the concentration of 0–9.5 mg/g, and the topical administration of kojic acid at 50 μg/mL inhibits browning of the skin [33]. Although some researchers have reported the occurrence of thyroid adenomas when continuously administered, it is generally recognized that the consumption of kojic acid at levels normally found in food is not a concern for safety.

## 5. Oligosaccharides

Rice, barley, wheat, and soy contain various oligosaccharides consisting of glucose, fructose, xylulose, galactose, maltose, and arabinose [34]. For example, starch consists of α-1,4-linked glucose polymers, with α-1,6-linked branches [35]. Xylan is a polymer consisting of β-1,4-linked xylulose [36]. The genome of *A. oryzae* contains abundant genes of glycolytic enzymes [37]. For example, α-amylase degrades starch and generates various lengths of α-1,4-linked glucose polymers randomly with α-1,6-linked branches [38]. These glucose oligomers are degraded by pancreatic amylases to glucose, and the resultant glucose is absorbed in the small intestine [39]. However, since pancreatic enzymes do not degrade oligosaccharides which contain unique structures derived from starch or includes xylulose or arabinose, such oligosaccharides are not degraded in the small intestine, instead passing through to the large intestine and, as a result, are assimilated by intestinal microbes. Indeed, *koji* manufactured from rice and sake manufactured using *koji* contain various oligosaccharides including isomalto-pligosaccharide containing α-1,2-bond [40] and gluco-oligosaccharide with adjacent α-1,6 branches at the non-reducing end derived from starch (Figure 4) [41] and shoyu manufactured from rice koji and soy contains oligosaccharides consisting of galactose, arabinose, and galacturonic acid [42]. It has been reported that the dietary ingestion of oligosaccharides increases intestinal *Bifidobacterium* [43]. Moreover, the dietary ingestion of oligosaccharide is beneficial for ulcerative colitis and minimal hepatic encephalopathy [44,45]. Furthermore, Shoyu polysaccharide produced by *koji* enzymes from soy alleviates perennial allergic rhinitis in humans [43] and promotes iron absorption in rats and humans [46].

*Koji*-fermented barley also contains oligosaccharides consisting of glucose, xylose, and arabinose that stimulate the growth of lactic acid bacteria and *Bifidobacterium* [47]. Since *Bifidobacterium infantis* ameliorates intestinal mucositis in rats [48] and *Bifidobacterium adolescentis* extract has an anti-proliferative effect on human colon cancer cell lines [49], it is considered that the dietary ingestion of *koji*-fermented foods and drinks (especially those manufactured from raw rice, barley, wheat, and soy) has potential medical applications to treat such patients.

## 6. Ethyl-α-d-Glucoside

Ethyl-α-d-glucoside (Figure 5), as well as α-glyceryl glucoside, ethyl-α-maltoside, and ethyl-α-isomaltoside, are formed from ethanol or glycerol and glucose, maltose, or isomaltose by the action of the transglycosidase produced by *koji* [50,51]. For example, the concentration of ethyl-α-d-glucoside is reported to be 0.33% *w*/*w* [50]. The dietary intake of the ethyl-α-d-glucoside (1% *w*/*w*) contained in sake and sake lees, the remnant of sake production, improves epidermal permeability barrier disruption by UVB irradiation in hairless mice [52]. It also upregulates collagen I and fibroblast growth factor I and VII in cultured human dermal fibroblasts [53]. Moreover, the topical application of ethyl-α-d-glucoside increases the intercellular lipid content, accelerates the differentiation of corneocytes, and reduces the thickness of the epidermis, thus improving the functions of the stratum corneum in murine epidermis [54]. Therefore, the dietary intake of sake or sake lees leads to the intake of ethyl-α-d-glucoside, thus resulting in improvement of epidermal permeability barrier disruption.

## 7. Ferulic Acid

Ferulic acid is ester-bound to carbohydrates (Figure 6) in cereals [55], and is formed by *koji* enzymes during sake brewing [56]. Ferulic acid is contained in sake at a concentration of 1–6 mg/L [57]. The dietary intake of ferulic acid lowers the lipid levels in hyperlipidemic diabetic rats [58]. The oral administration of 30 mg/kg/day ferulic acid or ethyl ferulate provides protective effects toward retinal degeneration [59]. Furthermore, the dietary intake of ferulic acid is protective toward the toxicity of the β-amyloid peptide, which causes Alzheimer’s disease [60]. Therefore, the dietary intake of *koji* leads to the intake of ferulic acids, thus resulting in amelioration of dyslipidemia, retinal degeneration, and Alzheimer’s disease.

## 8. Ergothioneine

Ergothioneine is a derivative of amino acid that has a strong antioxidative activity (Figure 7) [61,62]. *Koji* fermented with *A. oryzae* is reported to contain 59.8 ± 20.4 μg/g ergothioneine [62]. Feeding of ergothioneine at a concentration of 8 mg/kg body weight to mice protected against cisplatin-induced neuronal injury and enhances cognition [63].

Since skin cells and tissue can absorb ergothioneine, and it is accumulated in peripheral skin cells, it is considered to contribute to the antioxidant activity of skin cells and tissue [64].

Functional variants of the ergothioneine transporter, OCTN cation transporter genes, are associated with ulcerative colitis, Crohn’s disease, and autoimmune thyroid disease [65]. Therefore, it was hypothesized that the dietary intake of *koji* leads to the intake of ergothioneine and may contribute to the amelioration of these diseases.

## 9. Pyroglutamyl-Leucine

Peptides generated by proteases in *koji* are further modified during sake brewing (Figure 8). The pyroglutamyl-leucine contained in sake is one such example of a peptide, and is contained in sake at a concetration of 10–15 mg/L [66]. It has been shown to attenuate hepatitis and colitis in animal models [66].

## 10. Pyranonigrin A

It has been reported that *koji* contains some antioxidant substances [67,68]. Later, one such antioxidant substance was identified as pyranonigrin A (Figure 9) [69]. Pyranonigrin A is a potential main protease inhibitor of SARS-CoV-2 and might be utilized to prevent infection of coronavirus disease 2019 (COVID-19) [70]. Therefore, the dietary intake of *koji* and related fermented products leads to the intake of pyranonigrin A, resulting in the prevention of cardiovascular diseases and COVID-19.

## 11. Resistant Proteins

Most proteins in rice are degraded during sake brewing by the proteases contained in *koji*. However, nondigestable proteins, especially prolamin accumulated in the protein bodies of the endosperm of rice [71], remain in the sake lees, the remnant of sake brewing. Nondigestable proteins are called resistant proteins. As a result, sake lees contain a high content of resistant proteins. These resistant proteins are also resistant to human pancreatic proteases, and thus resistant proteins pass through the stomach and the small intestine and reaches the large intestine. As a result, they inhibit the absorption of lipids in the intestine, and improve intestinal microbial flora [72]. Therefore, the dietary intake of sake lees leads to the intake of resistant proteins, thus resulting in improvement of hyperlipidemia.

## 12. Deferriferrichrysin

*Koji* and its fungus *A. oryzae* contain a low-molecular-weight iron chelating peptide, deferriferrichrysin. Since it functions as a natural antioxidant [73], it might be beneficial for skin protection and prevention of inflammation.

## 13. Polyamines

During the manufacturing of *koji*, one of polyamines, agmatine (Figure 10), is produced at 3.1–8.7 mmol/L [74,75]. Polyamines have anti-inflammatory effects through the regulation of aberrant DNA methylation and decreased incidence of colon tumors, although accelerated the growth of established tumors [76]. Therefore, dietary intake of *koji* may lead to intake of polyamines, thus resulting in preventing the effects of colon tumors at a certain concentration.

## 14. *Bifidobacterium*-Stimulating Peptides

Proteases contained in *koji* acts on rice proteins and gives rise to various peptides. Certain peptides containing glutamate, serine, and alanine in *koji* were shown to increase *Bifidobacteria*, *B. longum, B. adolescentis,* and *B. breve* [77]. Therefore, *koji* is considered to improve the intestinal microbe.

## 15. Angiotensin I-Converting Enzyme Inhibitor Peptides

The renin–angiotensin system increases blood pressure. Angoitensin I-converting enzyme catalyzes the conversion of Ang I to Ang II, which eventually generates AT1R and causes proinflammatory effects [78,79]. Sake utilizing *koji* contains the angiotensin I-converting enzyme inhibitor peptides Val-Tyr, His-Tyr, Arg-Phe, Val-Trp, and Tyr-Trp. These peptides have been shown to be antihypertensive in spontaneously hypertensive rats [80,81]. Therefore, the dietary intake of sake and *koji*-fermented products leads to the intake of these peptides, thus resulting in anti-hypertensive effects.

## 16. 14-Dehydroergosterol

It was found that *Aspergillus luchuensis mut. kawachii* produces 14-dehydroergosterol (Figure 11) as an anti-inflammatory substance that induces tolerogenic dendritic cells [82]. Moreover, *koji* extract containing 14-dehydroergosterol produced by *Aspergillus luchuensis mut. kawachii* has been shown to improve skin moisture, TEWL, and skin wrinkles in humans in randomized, double-blind, controlled trial [83].

## 17. Beta-Glucan

Beta-glucan (Figure 12) was first identified as a medical component of *Agaricus blazeii* from the observations of lower incidence of cancers and infections of virus and bacteria, as well as the increased life span within native Americans [84]. Later, it was found that beta-glucan contained in the fungus is an immunostimulant and activates macrophages through dectin1 and CR3 (CD11b/CD18) [85,86]. It also ameliorates allergies and reduces the triglyceride and cholesterol levels in the blood [87]. *Aspergillus* spp. contains β-(1→3)-glucan as its cell wall component (Figure 11), and β-(1→3)-glucan of *Aspergillus fumigatus*, a close relative of *A. oryzae*, activates dectin-1 [88]. Since *koji* is manufactured using *A. oryzae*, it is considered to contain beta-glucan.

The administration of beta-glucan is considered to improve the quality of life of cancer patients and is useful as a complementary or adjuvant therapy and immunomodulatory agent in cancer patients in combination with other cancer treatments [89,90]. Therefore, the content of beta-glucan in *koji* is desired to be investigated.

## 18. Biotin

Lack of biotin (Figure 13) is known to causes skin disorders such as dermatitis, hair loss, neuritis, and susceptibility to infections [91]. It is also severely teratogenic in rodents. Since *Aspergillus oryzae* used for production of *koji* has a biosynthetic pathway of biotin [92] and *koji* contains 1.01 mg/100 g biotin [93], dietary intake of *koji* might contribute to the prevention of these disorders.

## 19. Citric Acid

*A. luchuensis*, which has been used for manufacturing awamori and shochu in Okinawa prefecture and Kyusyu island, produces citric acid (Figure 14) and koji fermented with *A. luchuensis* contains citric acid at approximately 1800 mg/L [94]. Therefore, *koji* manufactured using *A. luchuensis* contains citric acid (Figure 12). The dietary intake of citric acid (2 g/L) in drinking water inhibits the development of cataracts, proteinuria, and ketosis in diabetic rats [95]. Moreover, in a mouse model of systemic inflammation, dietary intake of citric acid (1–2 g/kg) has been shown to decrease brain lipid peroxidation and inflammation, liver damage, and DNA fragmentation in lipopolysaccharide-treated mice [96]. Therefore, fermented foods containing *koji* manufactured with *A. luchuensis* might have beneficial health effects toward diabetic patients and chronic inflammation, which should be a target of future study.

## 20. Potential Risk of *Koji*

*A. oryzae* in general produces aflatoxin and cyclopiazonic acid [97]. Ingestion of aflatoxin causes an acute hepatic necrosis [98]. Cyclopiazonic acid shows immunosuppressive activity at low doses and causes necrosis of various tissues [99]. However, since *A. oryzae* used for manufacture of *koji* has been long succeeded in Japan, it has lost functional genes to produce aflatoxin [100] or acquired a new gene to produce less toxic cyclopiazonic acid [101]. Also, *A. luchuensis* has lost the functional gene to synthesize ochratoxin A [102].

It should be noted that some strains produce trace amounts of certain mycotoxins including aspirochlorin and csypyrones [97] and it is necessary to monitor the concentrations of these substances.

## 21. Conclusions

In conclusion, *koji* fermented with *A. oryzae* or *A. luchuensis* and *koji*-fermented products contain abundant substances that have the potential for use in medical applications (Figure 15). It is desired that the medical benefits of these substances be verified in human studies and applied to practical medical use. More research is desired to medically apply *koji* to cure diseases across the world in the future.

## Figures and Tables

**Figure 1 jof-07-00243-f001:**
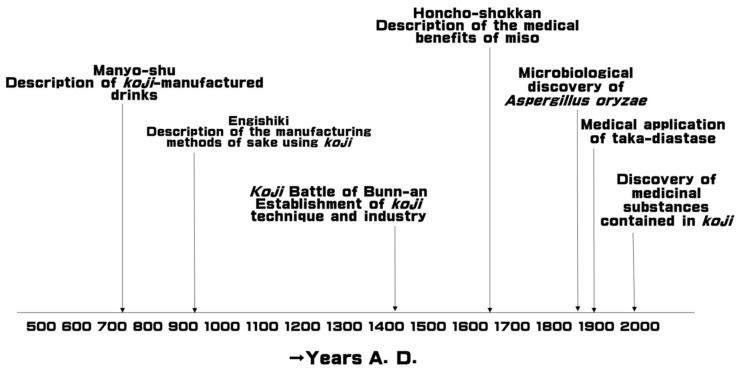
History of technologies and research on *koji* in Japan.

**Figure 2 jof-07-00243-f002:**
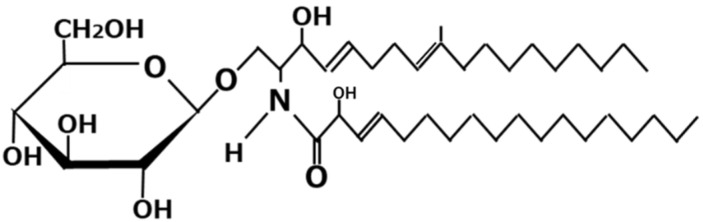
Chemical structure of glycosylceramide.

**Figure 3 jof-07-00243-f003:**
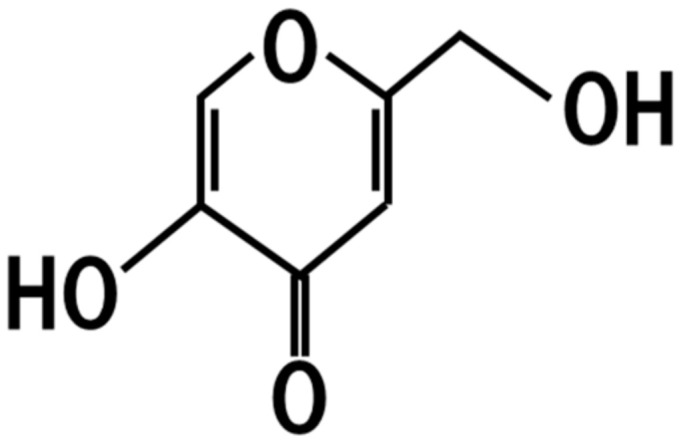
Chemical structure of Kojic acid.

**Figure 4 jof-07-00243-f004:**
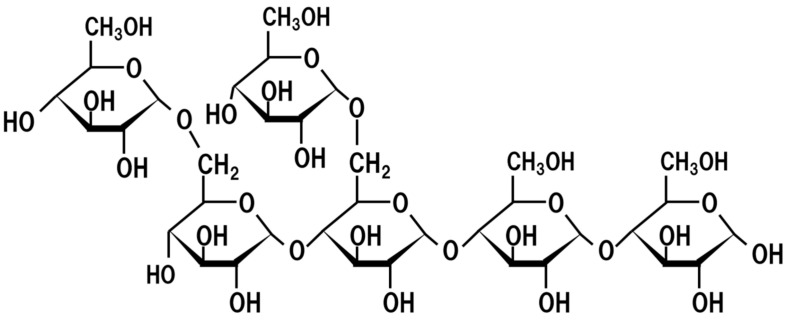
Chemical structure of resistant gluco-oligosaccaride.

**Figure 5 jof-07-00243-f005:**
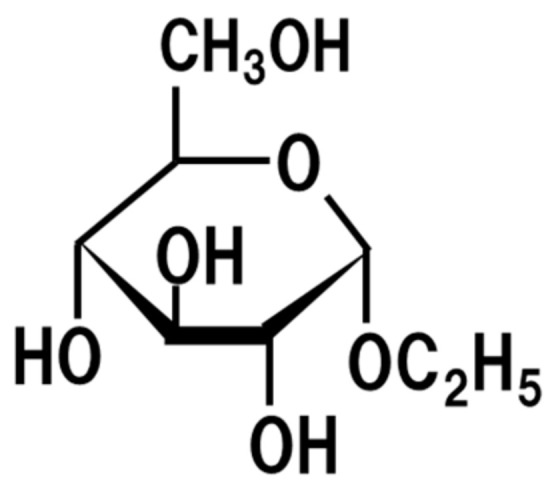
Chemical structure of ethyl-α-d-glucoside.

**Figure 6 jof-07-00243-f006:**
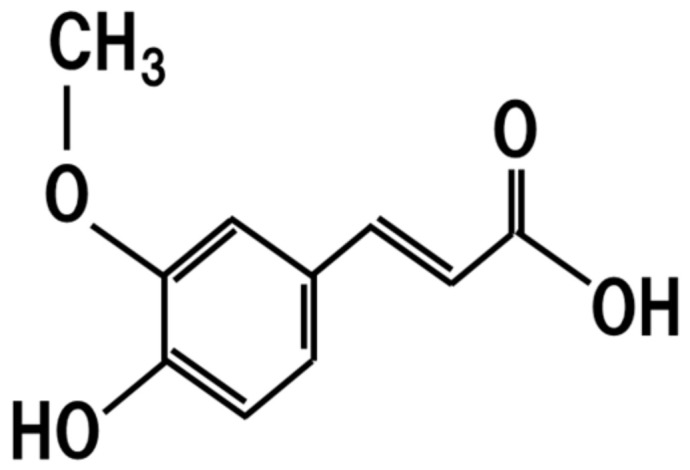
Chemical structure of ferulic acid.

**Figure 7 jof-07-00243-f007:**
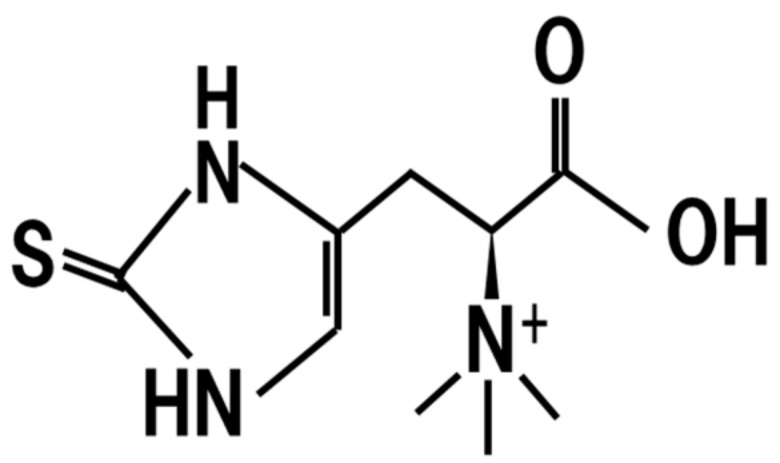
Chemical structure of ergothioneine.

**Figure 8 jof-07-00243-f008:**
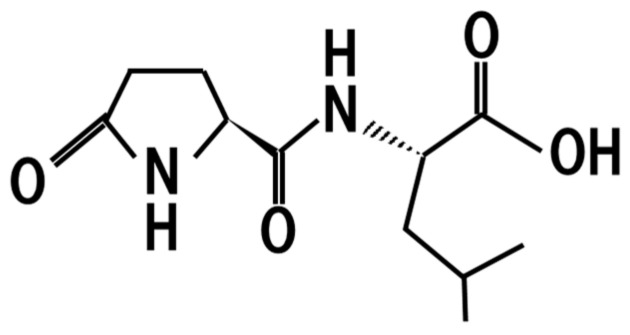
Chemical structure of pyroglutamyl-leucine.

**Figure 9 jof-07-00243-f009:**
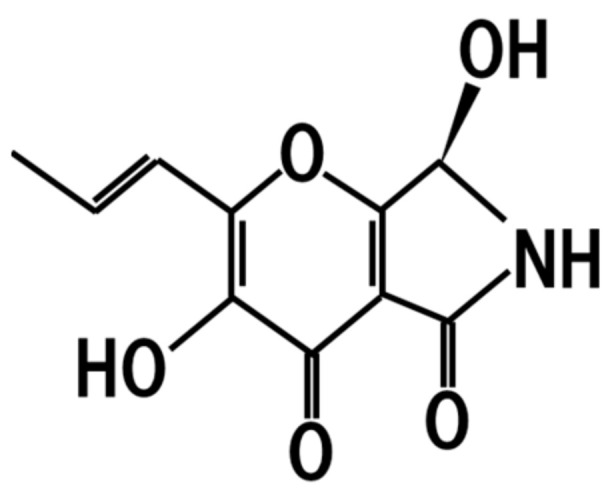
Chemical structure of pyranonigrin-A.

**Figure 10 jof-07-00243-f010:**
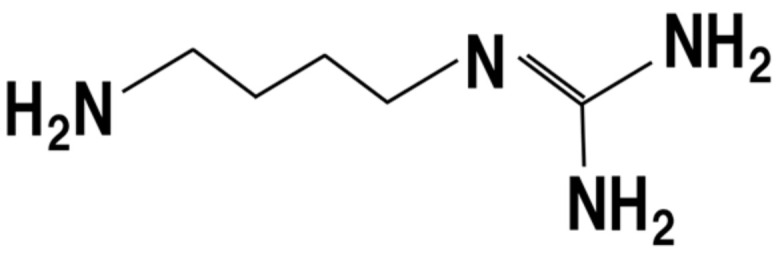
Chemical structure of agmatine.

**Figure 11 jof-07-00243-f011:**
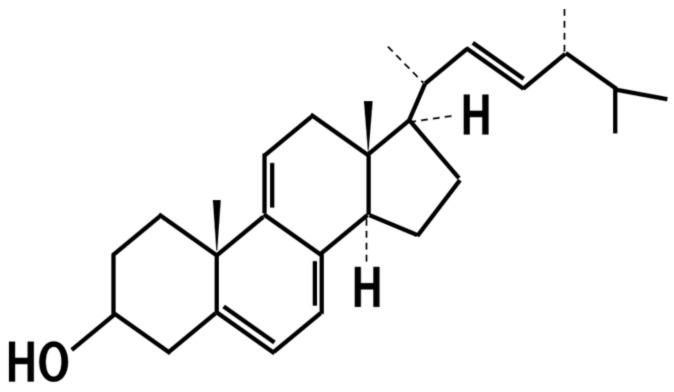
Chemical structure of 14-dehydroergosterol.

**Figure 12 jof-07-00243-f012:**
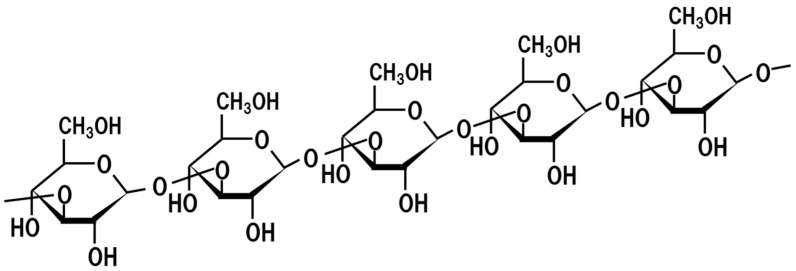
Chemical structure of beta-glucan.

**Figure 13 jof-07-00243-f013:**
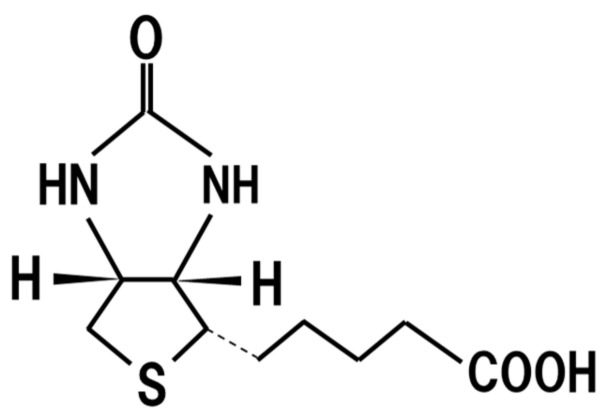
Chemical structure of biotin.

**Figure 14 jof-07-00243-f014:**
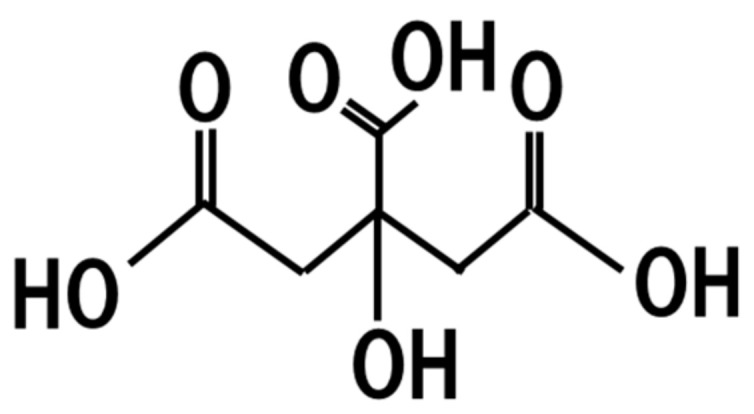
Chemical structure of citric acid.

**Figure 15 jof-07-00243-f015:**
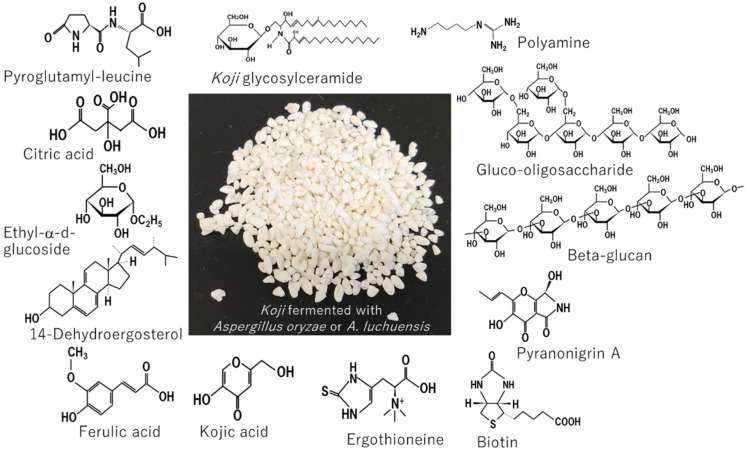
Medicinal substances contained in nonpathogenic *Aspergillus*-fermented *Koji*.

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
