# Peer review of "Medical Application of Substances Derived from Non-Pathogenic Fungi Aspergillus oryzae and A. luchuensis-Containing Koji"

_jof, 2021, doi:10.3390/jof7040243_

Round 1

Reviewer 1 Report

very short text for a review

the author has full legitimity in this scientific field (Scopus 70 refs H 21)

example

Article  •  Open Access

Glycosylceramides purified from the japanese traditional non-pathogenic fungus Aspergillus and Koji increase the expression of genes involved in tight junctions and ceramide delivery in normal human epidermal keratinocytes

Miyagawa, M.Fujikawa, A.Nagadome, M., ...Kitamura, S.Kitagaki, H. Fermentation20195(2)43

line 51: please explain why strains used for koji making are non-pathogenic (Aspergillus oryzae and Aspergillus luchuensis)

all strains of Aspergillus oryzae and Aspergillus luchuensis in the world are non pathogenic, non mycotoxin producers?

or Asia people selected some?

spell out the A. of A. luchuensis in title and line 51 (first occurrence in the text)

Very nice historical perspective about fungi bringing health benefits to humans

line 62-63  why so little knowledge about medical/medicinal properties of A. oryzae, after so many centuries?

figure and table could/may have a better aspect, more 'trendy' (color, font etc)

provide chemical structures of beneficial compounds, when these are available/known

paper is a bit too much oriented koji/soy, lack of comparison with other fungi-fermeted grains in other locations in the world

no adverse effects of polyamines? in humans?

difficult to understand how the compounds are sorted in this review. Not by chemical classes?

difficult also to have a clear idea about the concentration level of each compound in koji, enough to be active?

Author Response

Dear reviewer1,

I really appreciate your logical review.

>line 51: please explain why strains used for koji making are non-pathogenic (Aspergillus oryzae and Aspergillus luchuensis)

This is because these strains have been succeeded for more than 600 years, and only safe strains were maintained. This was described in L58 to cliarify.

>all strains of Aspergillus oryzae and Aspergillus luchuensis in the world are non pathogenic, non mycotoxin producers?

or Asia people selected some?

No. To clarify, L58 was added.

>spell out the A. of A. luchuensis in title and line 51 (first occurrence in the text)

This was done.

>Very nice historical perspective about fungi bringing health benefits to humans

line 62-63  why so little knowledge about medical/medicinal properties of A. oryzae, after so many centuries?

This takes many pages to describe. Japanese medical community was under the strong influence of China since its historical ages. Chinese medical books were already the modern level at the age of 200BC. All textbooks in Japanese universities were imported from China since 700AD. After 1650AD, Japanese doctors tried to import the european scientific revolution into the Japanese medical community.  This was why Japanese stuffs were forgotten as a medical substance. Since this historical explanation is out of the range of this review, I just wrote "because of its modest medicinal effect"(L72).

>figure and table could/may have a better aspect, more 'trendy' (color, font etc)

provide chemical structures of beneficial compounds, when these are available/known

These were added as figures.

>paper is a bit too much oriented koji/soy, lack of comparison with other fungi-fermeted grains in other locations in the world

According to the advice, I added tempeh in Indonesia.

>no adverse effects of polyamines? in humans?

This was added (L211).

>difficult to understand how the compounds are sorted in this review. Not by chemical classes?

To increase the significance and readability of the review, the substances were ordered according to the significance of substances. Although the order might be subjective, I believe it should increase impact of the review to the readers.

>difficult also to have a clear idea about the concentration level of each compound in koji, enough to be active?

Available information was added.

Reviewer 2 Report

In this manuscript, the author summarizes potentially beneficial molecules derived from Aspergillus oryzae. This review might be of interest to readers of the journal, but several points should be addressed before it can be further considered for publication.

1. Throughout the manuscript, the author should provide more detailed information on the molecules mentioned in the manuscript. For example, what enzymes are included in Taka-diastase? What are the representative examples of “various peptides (line 134)”? What are the representative examples of “resistant proteins”?

2. Please provide the chemical structures of the small molecules mentioned in the text in order that readers can easily follow the manuscript. Also, please describe how these molecules show their activity (e.g., what are the target molecules for them?), where appropriate.

3. The author should also discuss the safety issues related to A. oryzae (e.g., production of aspirochlorine).

Minor points:

1. Please carefully check the compound names used in the manuscript. For example, “kojic” in “kojic acid” should not be italicized, and “pyranonigrin-A” should be “pyranonigrin A.”

2. Line 29 – Not all mushrooms are basidiomycetes. Also, “Basidiomycetes” should not be italicized, and there is no need to capitalize the first letter. The same thing applies to “Ascomycetes.”

3. Line 45 – Please clarify the meaning of “health substances.”

Author Response

Dear reviewer2,

I really appreciate your logical review.

>In this manuscript, the author summarizes potentially beneficial molecules derived from Aspergillus oryzae. This review might be of interest to readers of the journal, but several points should be addressed before it can be further considered for publication.

1. Throughout the manuscript, the author should provide more detailed information on the molecules mentioned in the manuscript. For example, what enzymes are included in Taka-diastase? What are the representative examples of “various peptides (line 134)”? What are the representative examples of “resistant proteins”?
These were added
L78 starch-degrading enzyme
L214 certain peptides containing glutamate, serine and alanine in koji (Original article does not describe the details)
L195 prolamin

>2. Please provide the chemical structures of the small molecules mentioned in the text in order that readers can easily follow the manuscript.
These were added as figures.
>Also, please describe how these molecules show their activity (e.g., what are the target molecules for them?), where appropriate.
I am afraid that most of these studies have not yet elucidated the concrete target of the molecules. This is because human bodies are complex and the targets are broad.

>3. The author should also discuss the safety issues related to A. oryzae (e.g., production of aspirochlorine).
This was added (L261-8)

Minor points:

>1. Please carefully check the compound names used in the manuscript. For example, “kojic” in “kojic acid” should not be italicized, and “pyranonigrin-A” should be “pyranonigrin A.”

this was corrected.

>2. Line 29 – Not all mushrooms are basidiomycetes. Also, “Basidiomycetes” should not be italicized, and there is no need to capitalize the first letter. The same thing applies to “Ascomycetes.”

This was corrected. I apologize for my mistakes.
>3. Line 45 – Please clarify the meaning of “health substances.”

this was changed to medicinal substances.